

# 1 Identification of hydrological model parameters variation using

# 2 ensemble Kalman filter

Chao Deng[1,2], Pan Liu[1,2,*], Shenglian Guo[1,2], Zejun Li[1,2], Dingbao Wang[3]
[1]State Key Laboratory of Water Resources and Hydropower Engineering Science, Wuhan University,
Wuhan, China
[2]Hubei Provincial Collaborative Innovation Center for Water Resources Security, Wuhan, China
[3]Department of Civil, Environmental & Construction Engineering, University of Central Florida,
Orlando, USA
*Corresponding author: P. Liu, State Key Laboratory of Water Resources and Hydropower
Engineering Science, Wuhan University, Wuhan 430072, China
Email: liupan@whu.edu.cn
Tel: +86-27-68775788; Fax: +86-27-68773568


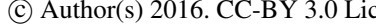



**Abstract**: Hydrological model parameters play an important role in the ability of model prediction. In
a stationary content, parameters of hydrological models are treated as constants. However, model
parameters may vary dynamically with time under climate change and human activities. The technique
of ensemble Kalman filter (EnKF) is proposed to identify the temporal variation of parameters for a
two-parameter monthly water balance model by assimilating the runoff observations, where one of state
equations is that the model parameters should not change much within a short time period. Through a
synthetic experiment, the proposed method is evaluated with various types of parameter variations
including trend, abrupt change, and periodicity. The application of the method to the Wudinghe basin
shows that the water storage capacity, a parameter in the model, has an apparent increasing trend during
the period from 1958 to 2000. The identified temporal variation of water storage capacity is explained
by land use and land cover changes due to soil and water conservation measurements. Whereas, the
application to the Tongtianhe basin demonstrates that the parameter of water storage capacity has no
significant variation during the simulation of 1982-2013, corresponding to the relatively stationary
catchment characteristics. Additionally, the proposed method improves the performance of hydrological
modeling, and provides an effective tool for quantifying temporal variation of model parameters.
**Keywords**: model parameter identification, temporal variation of parameter, catchment characteristics,
ensemble Kalman filter



# 1 Introduction

Hydrological model parameters are critically important for accurate simulation of streamflow. In hydrological modeling, parameters are usually assumed to be stationary, i.e., the calibrated parameters are a set of constants during the calibration period, and have extrapolative ability outside the range of the observations used for parameter estimation (Merz et al., 2011). However, the calibration period may contain different climactic condition and hydrological regime, and has a significant impact on the model parameter estimation (Merz et al., 2011; Zhang et al., 2011; Westra et al., 2014; Patil and Stieglitz, 2015). The model parameters may potentially change responding to time-variable precipitation and other inputs. For example, land use and land cover changes contribute to temporal change of model parameters (Andréassian et al., 2003; Brown et al., 2005; Merz et al., 2011). Consequently, assuming time invariant model parameters may be unrealistic, especially for catchments with time-varying climate conditions and/or catchment properties.

The situation of time-variant hydrological model parameters has been reported in a few publications (Merz et al., 2011; Brigode et al., 2013; Westra et al., 2014; Patil and Stieglitz, 2015). For example, Ye et al. (1997) and Paik et al. (2005) mentioned the seasonal variations of hydrological model parameters. Merz et al. (2011) analyzed the temporal changes of model parameters which were calibrated respectively by using six consecutive 5 year periods between 1976 and 2006 for 273 catchments in



Austria. Recently, Westra et al. (2014) proposed a strategy to cope with nonstationary of hydrological
model parameters, which were represented as a function of a set of time-varying covariates before using
an optimization algorithm for calibration. Previous studies provided two main methods to identify the
time-variant model parameters: (1) Divide the historical record into consecutive subsets, and then
calibrate the model parameters using an optimization algorithm (e.g., Merz et al. (2011)). The model
parameters are fixed values in each subset. (2) Build the functional form of the selected time-variant
model parameters, and calibrate the model parameters using an optimization algorithm based on the
entire historical record (e.g., Westra et al. (2014)).

The data assimilation (DA) method has been used to estimate both model parameters and state variables.
For example, Vrugt et al. (2013) proposed two types of Particle-DREAM method to track the evolving
target distribution of model parameters. Although the DA method has been used to estimate model
parameters, the objective is to identify the fixed values of parameters. Additionally, little attention has
been paid to the identification of time-variant model parameters and the interpretation of their temporal
variations based on the climate conditions and/or catchment characteristics.

The aim of this study is to assess the capability of the DA method (i.e., the EnKF) to identify the
temporal variation of parameters for a monthly water balance model, and to link the parameter



variations to changes in physical properties.

The remainder of this paper is organized as follows. Section 2 presents a brief review of the
two-parameter monthly water balance model and the EnKF method. Following the methodology,
Section 3 describes the synthetic experiment and the application to two case studies. Results and
discussion are presented in Section 4, followed by conclusions in Section 5.

## 2 Methodology

### 2.1 Monthly water balance model

The two-parameter monthly water balance model, developed by Xiong and Guo (1999), has been
widely applied for monthly runoff simulation and forecast (Guo et al., 2002; Guo et al., 2005; Xiong
and Guo, 2012; Li et al., 2013; Zhang et al., 2013; Xiong et al., 2014). The inputs of the model include
monthly areal precipitation and potential evapotranspiration. The actual monthly evapotranspiration is
calculated as follows:
$$E_i = C \times EP_i \times \tanh\left(P_i / EP_i\right) \tag{1}$$
where $E_i$ represents the actual monthly evapotranspiration; $EP_i$ and $P_i$ are the monthly potential
evapotranspiration and precipitation, respectively; $C$ is the first model parameter; and $i$ is the time
step.




The monthly runoff is dependent on the soil water content and is calculated by the following formula:
$Q_i = S_i \times \tanh\left(S_i / SC\right)$ (2)
where $Q_i$ is the monthly runoff; and $S_i$ is the soil water content. As the second model parameter,
$SC$ represents the water storage capacity of the catchment with the unit of millimeter. The available
water for runoff at the $i$ th month is computed by $S_{i-1} + P_i - E_i$. Then, the monthly runoff is calculated
by:
$Q_i = \left(S_{i-1} + P_i - E_i\right) \times \tanh\left[\left(S_{i-1} + P_i - E_i\right) / SC\right]$ (3)

Finally, the soil water content at the end of each time step is updated based on the water conservation
law:
$S_i = S_{i-1} + P_i - E_i - Q_i$ (4)

## 101 2.2 Ensemble Kalman filter

EnKF is a sequential data assimilation technique based on the Monte Carlo method and produces an
ensemble of state simulations to update the state variables and model parameters, conditioned on a
series of model observations (Moradkhani et al., 2005; Shi et al., 2014). It has been successfully
applied into dozens of hydrological applications (Abaza et al., 2014; DeChant and Moradkhani, 2014;



Delijani et al., 2014; Samuel et al., 2014; Tamura et al., 2014; Xue and Zhang, 2014; Deng et al.,
2015). In EnKF, the state equation is as follows:
$\theta_{i+1} = \theta_i + \delta_i, \delta_i \sim N(0, R_i)$ (5)
$x_{i+1} = f(x_i, \theta_{i+1}) + \varepsilon_i, \varepsilon_i \sim N(0, G_i)$ (6)
where $x_i$ is the state vector with a dimension of $n \times 1$ at time $i$; $\theta_{i+1}$ is the parameter vector with a
dimension of $l \times 1$ at time $i+1$; $f$ is the forward operator; $\varepsilon_i$ and $\delta_i$ are the independent white
noise for the forecast model with a dimension of $n \times 1$, followed a Gaussian distribution with zero
mean and covariance matrix $G_i$ and $R_i$ with a dimension of $n \times n$, respectively. Equation (5)
indicates that hydrological parameters should not change much within a short time period.

The observation equation is as follows:
$y_{i+1} = h(x_{i+1}, \theta_{i+1}) + \xi_{i+1}, \xi_{i+1} \sim N(0, S_{i+1})$ (7)
where $y_{i+1}$ is the observation vector with a dimension of $m \times 1$ at time $i+1$; $h$ is the
observational operator which represents the relationship between the observations and states; $\xi_{i+1}$ is
the noise term with a dimension of $m \times 1$ which follows a Gaussian distribution with zero mean and
covariance matrix $S_{i+1}$ with a dimension of $m \times m$.

Based on the available state and observation equations, the EnKF assimilation process can be





expressed as follows:
(1) Set the ensemble size $N$ and the total length of the historical record $n$.
(2) Generate the ensemble of model parameters and state variables by perturbing the updated values
from the previous time step.
$\theta_{i+1|i}^k = \theta_{i|i}^k + \delta_i^k$ (8)
$x_{i+1|i}^k = f\left(x_{i|i}^k, \theta_{i+1|i}^k\right) + \varepsilon_i^k$ (9)
where $\theta_{i+1|i}^k$ is the $k$th ensemble member forecast at time $i+1$; $\theta_{i|i}^k$ is the $k$th updated ensemble
member at time $i$; $\delta_i$ is the white noise for the $k$th ensemble member; $x_{i+1|i}^k$ is the $k$th ensemble
member forecast at time $i+1$; $x_{i|i}^k$ is the $k$th updated ensemble member at time $i$; and $\varepsilon_i^k$ is the
white noise for the $k$th ensemble member.
(3) Generate the ensemble of runoff observations by adding a perturbation:
$y_{i+1}^k = y_{i+1} + \xi_{i+1}^k$ (10)
where $y_{i+1}^k$ is the $k$th observation ensemble member at time $i+1$; and $\xi_{i+1}^k$ is the observation error
for the $k$th ensemble member.

The model parameters and state variables are updated according to the following equations:
$x_{i+1|i+1}^k = x_{i+1|i}^k + K_{i+1}^x\left(y_{i+1}^k - h\left(x_{i+1|i}^k, \theta_{i+1|i}^k\right)\right)$ (11)
$\theta_{i+1|i+1}^k = \theta_{i+1|i}^k + K_{i+1}^\theta\left(y_{i+1}^k - h\left(x_{i+1|i}^k, \theta_{i+1|i}^k\right)\right)$ (12)






Note that the parameter and state vectors are updated following the approach in the previous studies
(Wang et al., 2009; Nie et al., 2011; DeChant and Moradkhani, 2012; Lü et al., 2013). $K_{i+1}$ is the
Kalman gain matrix that represents the weight between the forecasts and observations. It can be
calculated by (Moradkhani et al., 2005):
$$K_{i+1}^{x} = \sum_{i+1|i}^{xy} \left( \sum_{i+1|i}^{yy} + S_{i+1} \right)^{-1} \tag{13}$$
$$K_{i+1}^{\theta} = \sum_{i+1|i}^{\theta y} \left( \sum_{i+1|i}^{yy} + S_{i+1} \right)^{-1} \tag{14}$$
$$\sum_{i+1|i}^{xy} = \frac{1}{N-1} X_{i+1|i} Y_{i+1|i}^{T} \tag{15}$$
$$\sum_{i+1|i}^{\theta y} = \frac{1}{N-1} \Theta_{i+1|i} Y_{i+1|i}^{T} \tag{16}$$
$$\sum_{i+1|i}^{yy} = \frac{1}{N-1} Y_{i+1|i} Y_{i+1|i}^{T} \tag{17}$$
where $\sum_{i+1|i}^{xy}$ is the cross covariance of the forecasted states; $\sum_{i+1|i}^{\theta y}$ is the cross covariance of the forecasted
parameters; $\sum_{i+1|i}^{yy}$ is the error covariance of the forecasted output; $X_{i+1|i} = \left( x_{i+1|i}^{1} - x_{i+1|i}^{m}, \cdots, x_{i+1|i}^{N} - x_{i+1|i}^{m} \right)$ and $x_{i+1|i}^{m}$
is the ensemble mean of the forecasted states; $\Theta_{i+1|i} = \left( \theta_{i+1|i}^{1} - \theta_{i+1|i}^{m}, \cdots, \theta_{i+1|i}^{N} - \theta_{i+1|i}^{m} \right)$ and $\theta_{i+1|i}^{m}$ is the ensemble mean
of the forecasted parameters; $Y_{i+1|i} = \left( y_{i+1|i}^{1} - y_{i+1|i}^{m}, \cdots, y_{i+1|i}^{N} - y_{i+1|i}^{m} \right)$ and $y_{i+1|i}^{m}$ is the ensemble mean of the
forecasted output; $N$ is the number of ensemble members; and the superscript $T$ represents the matrix transpose.
Since the parameters are limited within an interval, the constrained EnKF is used (Wang et al., 2009) in this study.

The ensemble size, uncertainties in input and output have significant impacts on the assimilation





performance of the EnKF, and they are determined based on previous studies (Wang et al., 2009; Xie
and Zhang, 2010; Lü et al., 2013; Samuel et al., 2014). The ensemble size is set to 1000 for all cases.
In the present study, the uncertainties including parameter errors ($\varepsilon$, Eq. (8)), state variable error ($\delta$,
Eq. (9)) and streamflow observation error ($\xi$, Eq. (10)), are assumed to follow a Gaussian distribution.
In terms of the parameter errors, the standard deviation for $C$ is set to 0.01 for all the cases, while
that of $SC$ are set from 0.5 to 5 to account for its uncertainties. The standard deviation of both model
state and observation errors are assumed to be proportional to the magnitude of true values, and the
scale factors are set to be 5% and 10% respectively for all cases (Wang et al., 2009; Lü et al., 2013). It
should be noted that the variable variance multiplier can be used to perturb the observations
(Leisenring and Moradkhani, 2012; Yan et al., 2015).

**2.3 Evaluation index**
Two evaluation criteria, including the Nash-Sutcliffe efficiency (NSE) (Nash and Sutcliffe, 1970) and
the volume error (VE) are used to evaluate the runoff assimilation results for the synthetic experiment
and application to catchments (Deng et al., 2015; Li et al., 2015).
$$NSE = 1 - \frac{\sum_{i=1}^{n}\left(Q_{sim,i} - Q_{obs,i}\right)^2}{\sum_{i=1}^{n}\left(Q_{obs,i} - \overline{Q}_{obs}\right)^2} \tag{18}$$





$$VE = \frac{\sum_{i=1}^{n} Q_{sim,i} - \sum_{i=1}^{n} Q_{obs,i}}{\sum_{i=1}^{n} Q_{obs,i}}$$ (19)
where $Q_{sim,i}$ and $Q_{obs,i}$ are the simulated and observed runoff for the $i$ th month; $\overline{Q}_{sim}$ and $\overline{Q}_{obs}$
are the mean of the simulated and observed runoff, respectively for the $i$ th month; and $n$ is the total
number of data points. The NSE has been widely used to assess the goodness-of-fit for hydrological
modeling. A NSE value of 1 means a perfect match of simulated runoff to the observations. The VE is
a measure of bias between the simulated and observed runoff. For example, VE with the value of 0
denotes no bias, and a negative value means an underestimation of the total runoff volume.

## 3 Data and study area

### 3.1 Synthetic experiment

A synthetic experiment is designed to evaluate the capability of the assimilation procedure to identify
the temporal variation of model parameters. The model parameters are given with specific variation
including trend, abrupt change and periodicity. Observations for precipitation and potential
evapotranspiration are generated via a stochastic simulation, and runoff is then produced by using the
monthly water balance model. The steps toward identifying temporal variation of model parameters are
as follows:
(1) Scenarios set: Generate the time-variant parameters with different trend variations, potential
evapotranspiration and precipitation on a monthly time scale, then compute the runoff observations





using the two-parameter monthly water balance model.
(2) Initialization: Specify the ensemble size and the total number of assimilation time steps. At the first
time step, the model parameter and state variable ensembles are generated using a predefined Gaussian
distribution based on the prior intervals in **Table 1**.
(3) Data assimilation: After the initialization of parameters and state variables, the hydrologic model
parameters and state are updated by assimilating the runoff observations obtained in Step (1). Note that
the model parameters and state, as well as the runoff observations, are perturbed with an error item
which is assumed a Gaussian distribution with zero mean and specified variance.

The data set used in this experiment has a total length of 672 months. The first 24 months is set as
model warm-up period to reduce the impact of the initial hydrological conditions. The experiment is
implemented to identify the variation of model parameters from the scenarios in **Table 2**, respectively.

The assimilated parameter results are evaluated using the following criteria, including the Pearson
correlation coefficient (R), the root mean square error (RMSE) and mean absolute relative error
(MARE):
$$R = \frac{\sum_{i=1}^{n}\left(x_{sim,i} - \overline{x}_{sim}\right)\left(x_{obs,i} - \overline{x}_{obs}\right)}{\sqrt{\sum_{i=1}^{n}\left(x_{sim,i} - \overline{x}_{sim}\right)^2\left(x_{obs,i} - \overline{x}_{obs}\right)^2}} \qquad (20)$$





$$RMSE = \sqrt{\frac{1}{n}\sum_{i=1}^{n}\left(x_{sim,i} - x_{obs,i}\right)^2} \tag{21}$$
$$MARE = \frac{1}{n}\sum_{i=1}^{n}\frac{\left|x_{sim,i} - x_{obs,i}\right|}{x_{obs,i}} \tag{22}$$
where $x_{sim,i}$ and $x_{obs,i}$ are the assimilated and observed model parameters for the $i$th month; $\bar{x}_{sim}$
and $\bar{x}_{obs}$ are the mean of the assimilated and observed model parameters, respectively for the $i$th
month; $n$ is the total number of data points.
**3.2 Study area**
**3.2.1 Case 1: Wudinghe basin**
The method is applied in the Wudinghe basin (**Fig. 1**) located in the southern fringe of Maowusu
Desert and the northern part of the Loess Plateau in China with a semiarid climate. It has a drainage
area of approximately 30,261 km² and a total length of 491 km and forms a part of the Yellow River
basin. The Wudinghe basin has an average slope of 0.2%, and its elevation ranges from 600 to 1800 m
above the sea level. The Baijiachuan gauge station, which is the most downstream station of the
Wudinghe basin, drains 98% of the total catchment area. The mean annual precipitation over the basin
is 401 mm, of which 72.5% occurs in the rainy season from June to September (**Fig. 2**). The mean
annual potential evapotranspiration is 1077 mm, and the mean annual runoff is about 39 mm with a
runoff coefficient of 0.1.


### 3.2.2 Case 2: Tongtianhe basin

The Tongtianhe basin (**Fig. 3**) is located in southwestern Qinghai Province in China with a continental climate. It belongs to the source area of Yangtze River basin with a drainage area of about 140,000 km$^2$ and a total main stream length of 1206 km. The elevation of the Tongtianhe basin approximately ranges from 3500 to 6500 m above the sea level. Zhimenda is the basin outlet. The mean annual precipitation over the basin is 440 mm, of which 76.9% occurs in the period from June to September (**Fig. 4**). The mean annual potential evapotranspiration is 796 mm, and the mean annual runoff is about 99 mm with a runoff coefficient of 0.23. The Tongtianhe basin is barely affected by human activities owing to the limitation of the topographic condition and the water conservation measures conducted by the government. It should be noted that the Tongtianhe basin is used for comparative study on model parameter identification, where has no significant impacts from the climate change and human activities.

### 3.2.3 Data

The data set including monthly precipitation, potential evapotranspiration and runoff in Wudinghe basin (from 1956 to 2000) and Tongtianhe basin (from 1980 to 2013) are used in this study. The potential evapotranspiration is estimated using the Penman-Monteith equation (Allen et al., 1998) based on the meteorological data from the China Meteorological Data Sharing Service System (http://cdc.nmic.cn).



To reduce the impact of the model initial conditions, a 2-year data set, i.e., from 1956 to 1957 for
Wudinghe basin and from 1980 to 1981 for Tongtianhe basin, is reserved as the warm-up period. The
runoff estimations from the SCE-UA method (Duan et al., 1993) are compared with that of the EnKF.

## 4 Results and discussion

### 4.1 Synthetic experiment

To assess the performance of the EnKF, the assimilated results are examined for the four scenarios in
the synthetic experiment. The comparisons of the assimilated and true model parameters under
different scenarios are presented from **Fig. 5** to **Fig. 8**, and **Table 3** shows the evaluation statistics for
both the parameters and runoff assimilations. All these four figures show that the assimilated
parameters of $C$ and $SC$ have similar trends as the true ones. These figures demonstrate that the
$SC$ assimilation performs better than the $C$ assimilation. The runoff assimilation results (see **Table**
**3**, penultimate and last columns) show that the estimation of runoff using the EnKF perfectly matches
the observations with NSEs of 0.99 and VEs of approximately zero. It should be noted that there is a
time lag in assimilated $C$ for the periodic case. In EnKF, the observation at the current time is used
to adjust the state variables and parameters, and the updates of parameters depend on the Kalman gain
for parameters.



The above results demonstrate that the EnKF is able to identify the temporal variation of the model
parameters by updating the state variable and parameters based on the runoff observations. The
estimated parameters for the cases of trend or abrupt change match the true values better than the case
with periodic variation.

**4.2 Case studies**
**Fig. 9** illustrates the double mass curve of monthly runoff and precipitation for Wudinghe and
Tongtianhe basins, respectively. The top panel shows the linear relationship between cumulative runoff
and precipitation before and after the turning point of January 1972 in the Wudinghe basin, which is
same as the result presented by Li et al. (2014). The results show two straight lines with different slopes
for the relationships between precipitation and runoff, indicating that changes occurred. While the
bottom panel demonstrates a single linear relationship fits all the data for the Tongtianhe basin,
suggesting a stable precipitation-runoff relationship during the 1982-2013 period.

The temporal variation of estimated $SC$ and the associated 95% uncertainty interval are shown in **Fig.**
**10**. The top panel shows an apparent increasing trend with two stages in Wudinghe basin. The first stage
is from January 1958 to December 1971, when the water storage capacity has a significant increasing
trend with a slope of 0.059. The water storage capacity in the second stage, from January 1972 to



December 2000, has an obvious increasing trend with a slope of 0.022. The temporal variation of water
storage capacity is related to the change of catchment properties, such as the land use and land cover
change. Since the 1960s, the soil and water conservation measures, including tree and grass planting,
reservoir construction and land terracing, have been undertaken to cope with the soil erosion in
Wudinghe basin. During the 1970s, large-scale engineering measures were effectively implemented,
which improved the water holding capacity of the basin directly, and also provided a reasonable
physical explanation for the increasing trend and its degree of $SC$ in the first stage. In the second stage,
the water storage capacity increases slower than the first stage since the engineering measures have
almost finished. Another important factor is the reduction of storage capacity for reservoirs caused by
sediment accumulation. In the 1980s, lots of measures were adopted for comprehensive management
within small catchments for further soil erosion control, which resulted in increasing grassland, forest
land and terracing land. These land use changes played a significant role in increasing water storage
capacity. On the other hand, the result of Tongtianhe basin shows that the estimated $SC$ has no
pronounced trend since the $R$ value has an insignificance level. Moreover, the range of variation in
estimated $SC$ values is much smaller compared to those of the Wudinghe basin. The grey regions
represent the 95% uncertainty intervals obtained from the parameter ensemble. The results demonstrate
that the EnKF performs well for parameter estimation with narrow uncertainty bounds. **Fig. 11** shows
the temporal variation of estimated $C$ values and the 95% uncertainty ranges for both Wudinghe basin



and Tongtianhe basin. The results demonstrate that the estimated $C$ has a stable value, with slopes that
are almost zero for both the cases. The narrow uncertainty bounds indicate that the EnKF can provide
superior performance of parameter estimation.

**Fig. 12** illustrates the comparison of the observed and estimated runoff from the EnKF and SCE-UA for
both Wudinghe and Tongtianhe basins. The evaluation results are shown in **Table 4**. The NSEs from the
EnKF and SCE-UA in the Wudinghe basin are 0.93 and 0.16, and the VEs are 0.07 and 0, respectively.
While the corresponding index values from the EnKF and SCE-UA are 0.99 and 0.79, 0.04 and 0 in the
Tongtianhe basin. Therefore, the EnKF has superior performance compared to the SCE-UA for both
case studies. The results show that the data assimilation improves the runoff estimation.

In summary, these analyses show that the EnKF can identify the temporal variation of model parameters
well by updating both state variables and parameters based on the runoff observations. Moreover, the
trends of parameter $SC$ can be explained by the change of catchment characteristics. On the contrary,
the estimated $SC$ is approximately stable when the catchment is barely affected by human activities.
Consequently, the EnKF provides effective performance for time-variant parameter identification.

## 317 5 Conclusions





This study proposes an ensemble Kalman filter (EnKF) to identify the temporal variation of model
parameters in a monthly water balance. A synthetic experiment, which contains four scenarios of model
parameter variation, is designed to demonstrate the ability of the EnKF for identifying the temporal
variation of the model parameters using the runoff observations. The main conclusions are drawn as
follows.

Based on EnKF, the variation of model parameters can be effectively identified by assimilating runoff
observations. The EnKF can provide accurate results for parameter identification even though slight
time lags exist when parameters have periodic variations.

Then, the EnKF method is applied to the Wudinghe basin in China, aiming to detect the temporal
variability of model parameters and to provide an explanation for the parameter variation from the
perspective of catchment property change. Meanwhile, a comparative study is implemented to
investigate the variation of model parameters in Tongtianhe basin where human activities barely exist.
The parameter of water storage capacity ($SC$) for the monthly water balance model shows a significant
increasing trend for the period of 1958 to 2000 in the Wudinghe basin. The soil and water conservation
measures, including tree and grass planting, reservoir building and land terracing, have been
implemented during 1958 to 2000, resulting in the increase of the water holding capacity of the basin,



which explains the increasing trends of $SC$. Moreover, the magnitudes of the engineering measures in
different time periods play an important role in the degree of increasing trend for $SC$. In the Tongtianhe
basin, the parameter $SC$ has no significant trend for the period of 1982 to 2013, which is consistent
with the relatively stationary catchment characteristics.

The method proposed in this paper provides an effective tool for the time-variant model parameters
identification. Future work will be focused on the influence of the correlations between/among model
parameters and performance comparison of multiple data assimilation methods.

## Acknowledgments

This study was supported by the Excellent Young Scientist Foundation of NSFC (51422907). The
authors would like to thank the editor and the anonymous reviewers for their comments that helped
improve the quality of the paper.

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




# Tables

**Table 1.** Description and prior ranges of the two parameters for the monthly water balance model.

| Parameters and state variable | | Description | Interval and unit |
|---|---|---|---|
| Parameter | $C$ | Evapotranspiration parameter | 0.2-2.0 (-) |
| | $SC$ | Catchment water storage capacity | 100-2000 (mm) |
| State variable | $S$ | Soil water content | mm |







**Table 2.** Scenarios of time-variant model parameters in the synthetic experiment.

| Scenario | Description |
|---|---|
| Scenario 1 | C has a periodic variation, and SC has an increasing trend |
| Scenario 2 | C has a periodic variation, and SC has an abrupt change |
| Scenario 3 | C has a periodic variation with an increasing trend, and SC has an increasing trend |
| Scenario 4 | C has a periodic variation with an increasing trend, and SC has an abrupt change |





**Table 3.** Performance statistics for parameter and runoff estimations in the synthetic experiment.

| Scenario | Parameter | RMSE | R | MARE | NSE (Runoff) | VE (Runoff) |
|---|---|---|---|---|---|---|
| Scenario 1 | C | 0.15 | 0.554 | 0.21 | 0.99 | 0.0007 |
| | SC | 182.87 | 0.987 | 0.03 | | |
| Scenario 2 | C | 0.16 | 0.633 | 0.19 | 0.99 | 0.0001 |
| | SC | 156.19 | 0.957 | 0.04 | | |
| Scenario 3 | C | 0.12 | 0.636 | 0.12 | 0.99 | -0.0012 |
| | SC | 180.27 | 0.992 | 0.03 | | |
| Scenario 4 | C | 0.12 | 0.695 | 0.12 | 0.99 | -0.0009 |
| | SC | 156.42 | 0.969 | 0.03 | | |







**Table 4.** Comparison of monthly runoff simulation performance between the optimization algorithm (SCE-UA) and the data assimilation method (EnKF) in Wudinghe basin within the period 1958-2000 and Tongtianhe basin within the period 1982-2013, respectively.

| Area | Method | NSE | VE |
|---|---|---|---|
| Wudinghe basin | SCE-UA | 0.16 | 0 |
| | EnKF | 0.93 | 0.07 |
| Tongtianhe basin | SCE-UA | 0.79 | 0 |
| | EnKF | 0.99 | 0.04 |



**Figures**

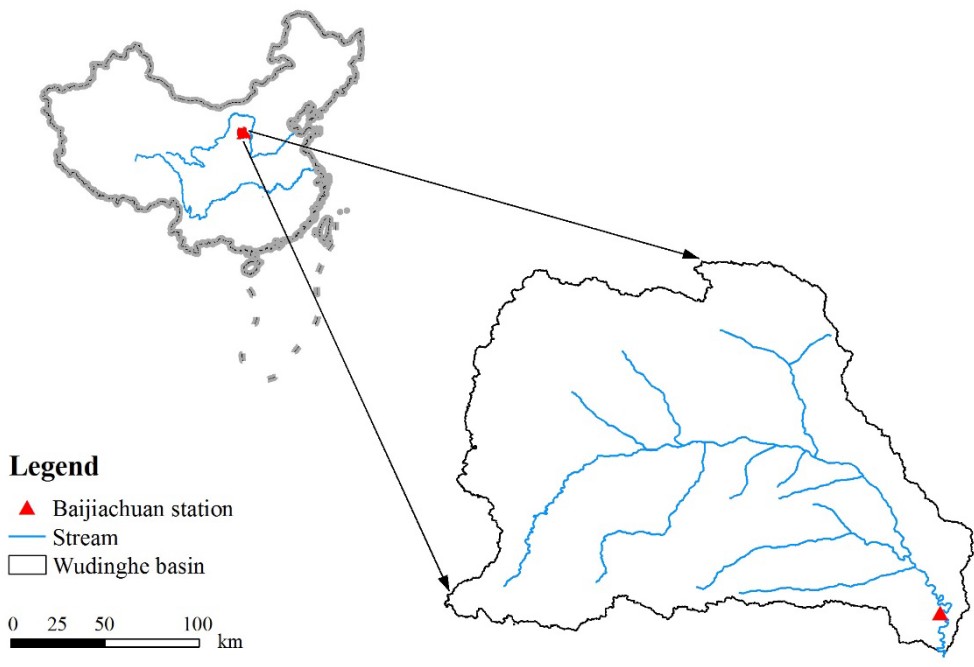


478                             **Figure. 1.** Location of Wudinghe basin.




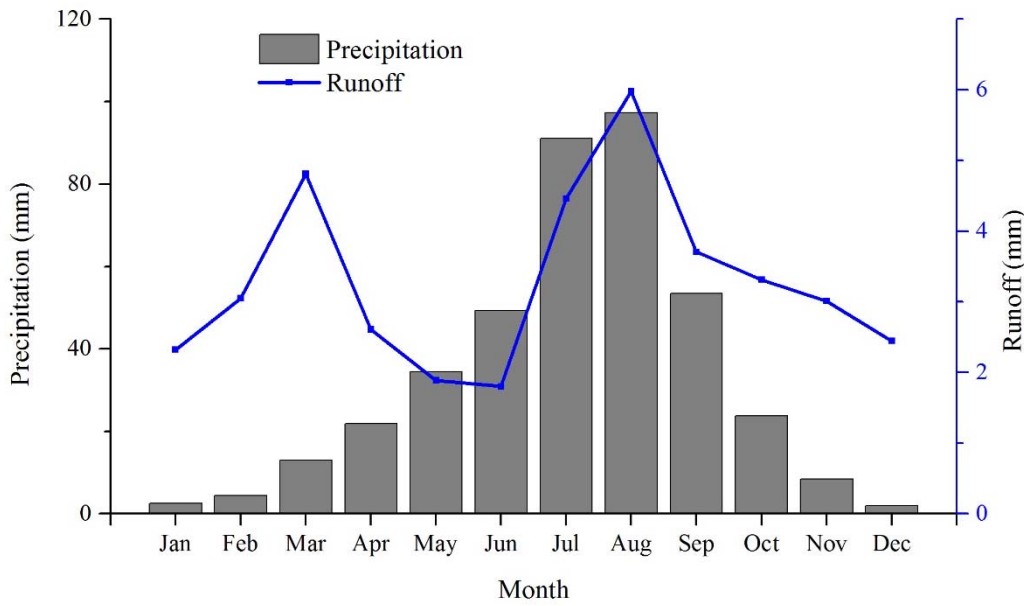

481       **Figure. 2.** Mean monthly precipitation and runoff from 1956 to 2000 in Wudinghe basin.




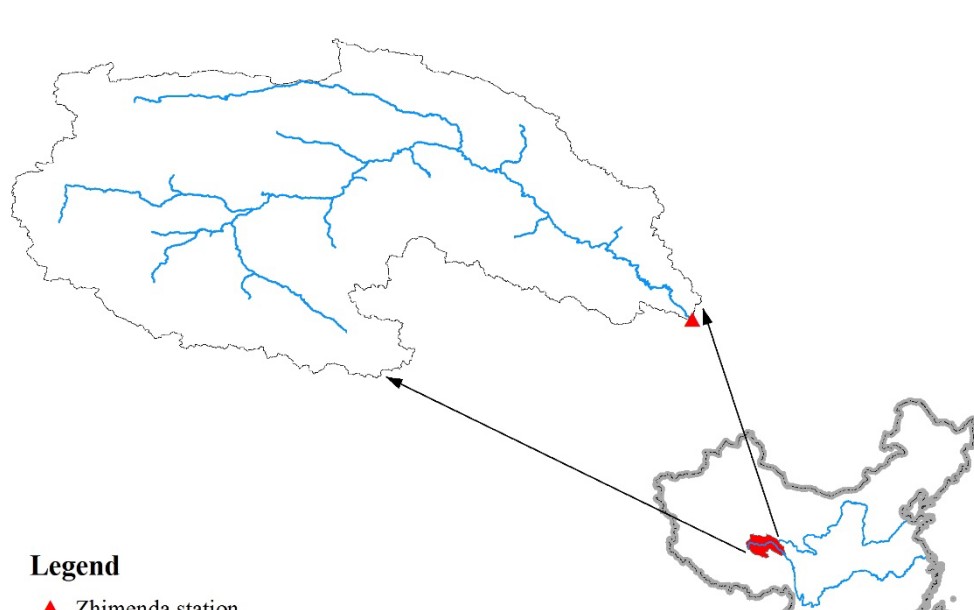


**Figure. 3.** Location of Tongtianhe basin.






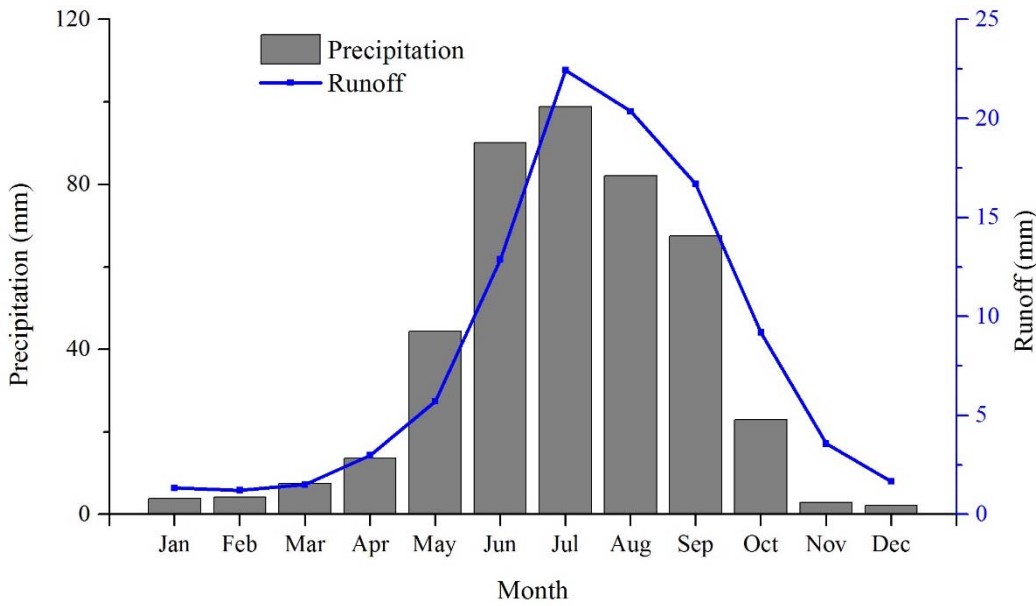


**Figure. 4.** Mean monthly precipitation and runoff from 1980 to 2013 in Tongtianhe basin.




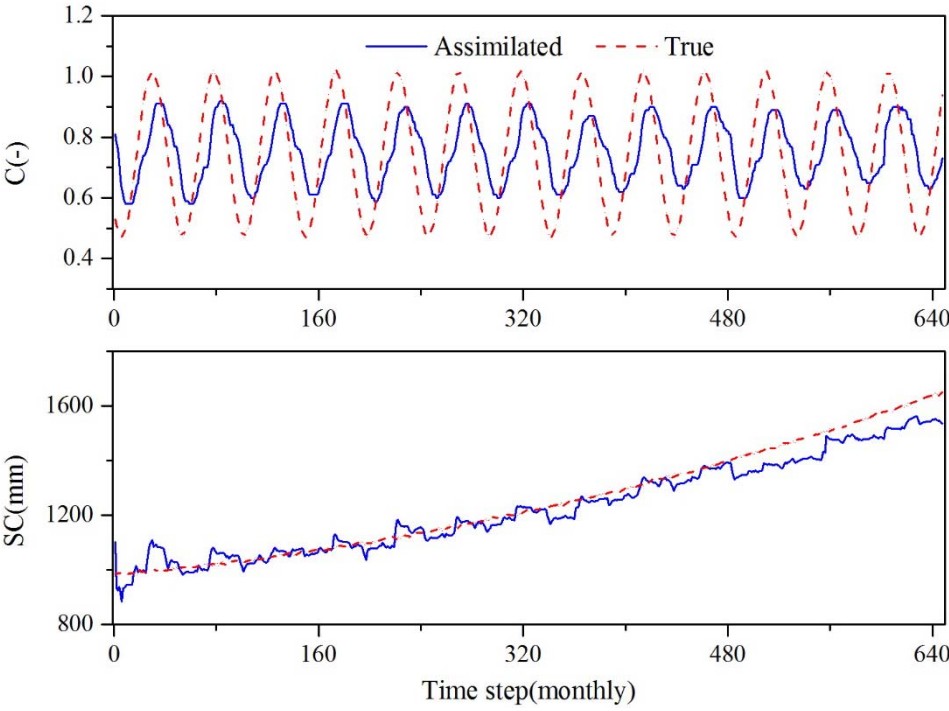


**Figure. 5.** Model parameters (evapotranspiration parameter C, water storage capacity SC) of assimilated and true in the synthetic experiment, considering C and SC are periodicity and increasing trend, respectively.




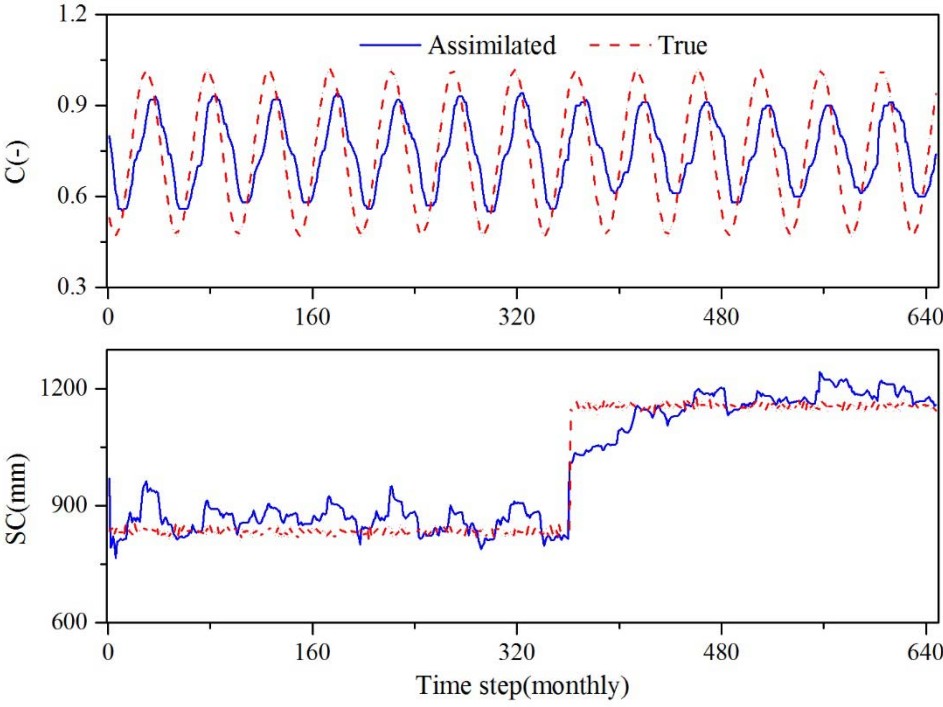


**Figure. 6.** Model parameters (evapotranspiration parameter C, water storage capacity SC) of assimilated and true in
the synthetic experiment, considering C and SC are periodicity and abrupt change, respectively.




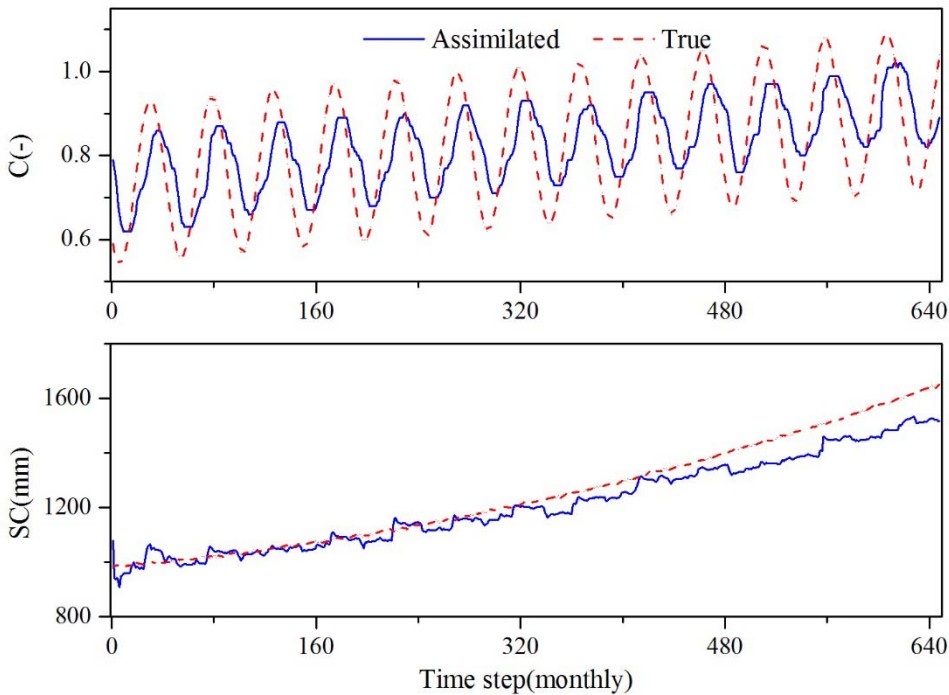


**Figure. 7.** Model parameters (evapotranspiration parameter C, water storage capacity SC) of assimilated and true in the synthetic experiment, considering C is periodicity with an increasing trend and SC is increasing trend, respectively.




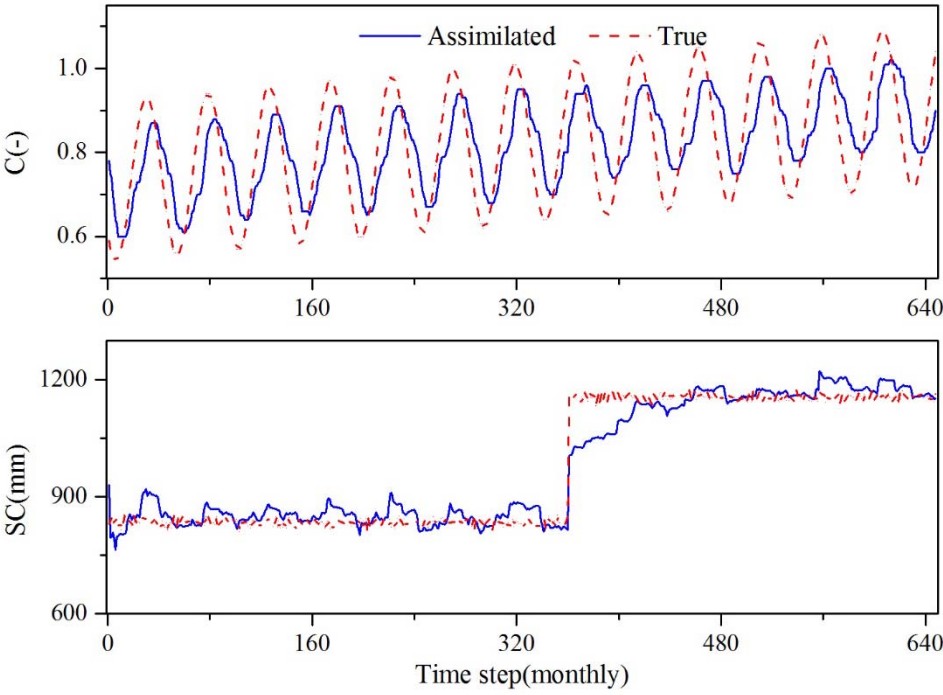


**Figure. 8.** Model parameters (evapotranspiration parameter C, water storage capacity SC) of assimilated and true in
the synthetic experiment, considering C is periodicity with an increasing trend and SC is abrupt change, respectively.






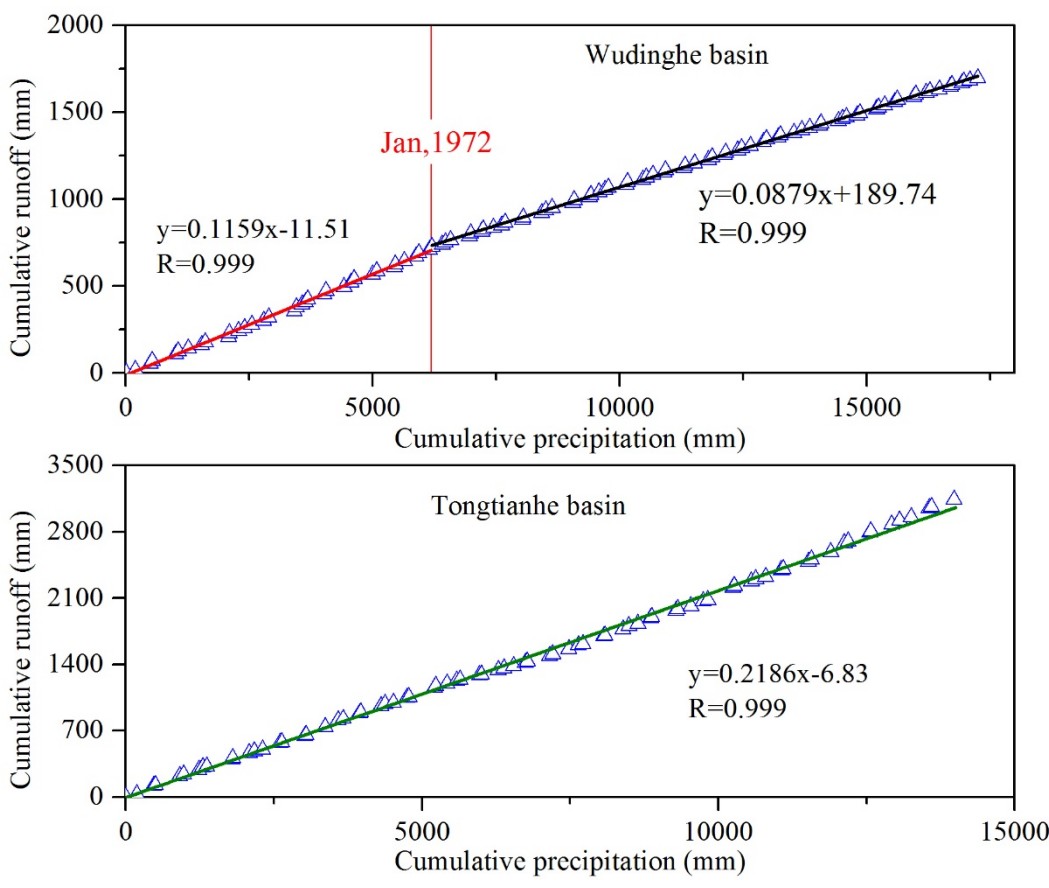

**Figure. 9.** Double mass curve of monthly runoff and precipitation for Wudinghe basin within the period 1958-2000
508             (top figure) and Tongtianhe basin within the period 1982-2013 (bottom), respectively.




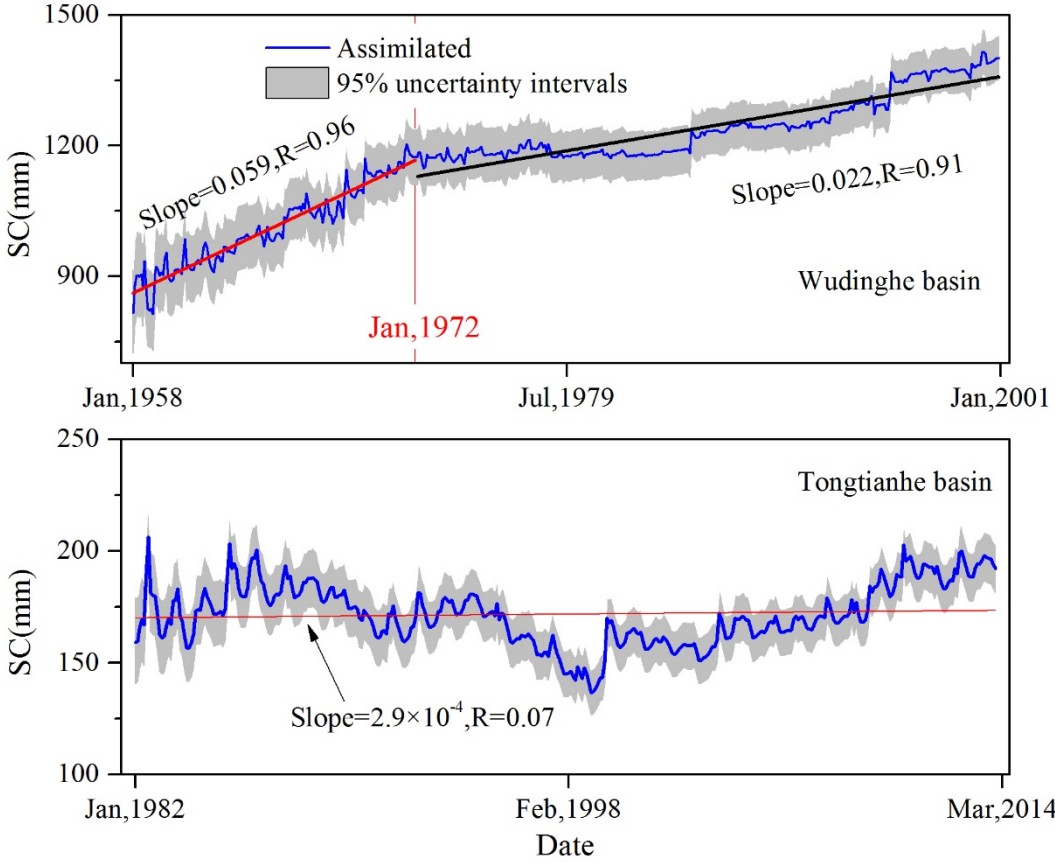


**Figure. 10.** Estimated parameter values of *SC* (water storage capacity) and associated 95% uncertainty intervals for Wudinghe basin within the period 1958-2000 (top figure) and Tongtianhe basin within the period 1982-2013 (bottom).



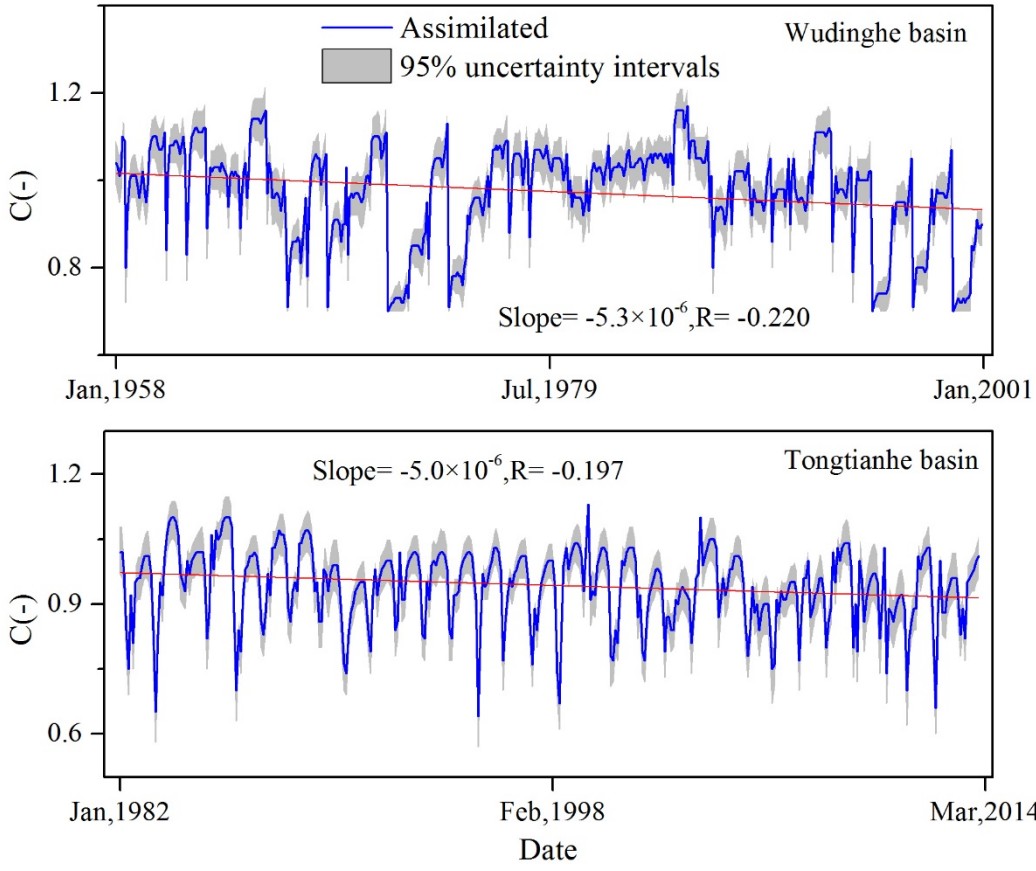


**Figure. 11.** Estimated parameter values of $C$ (evapotranspiration parameter) and associated 95% uncertainty intervals for Wudinghe basin within the period 1958-2000 (top figure) and Tongtianhe basin within the period 1982-2013 (bottom).




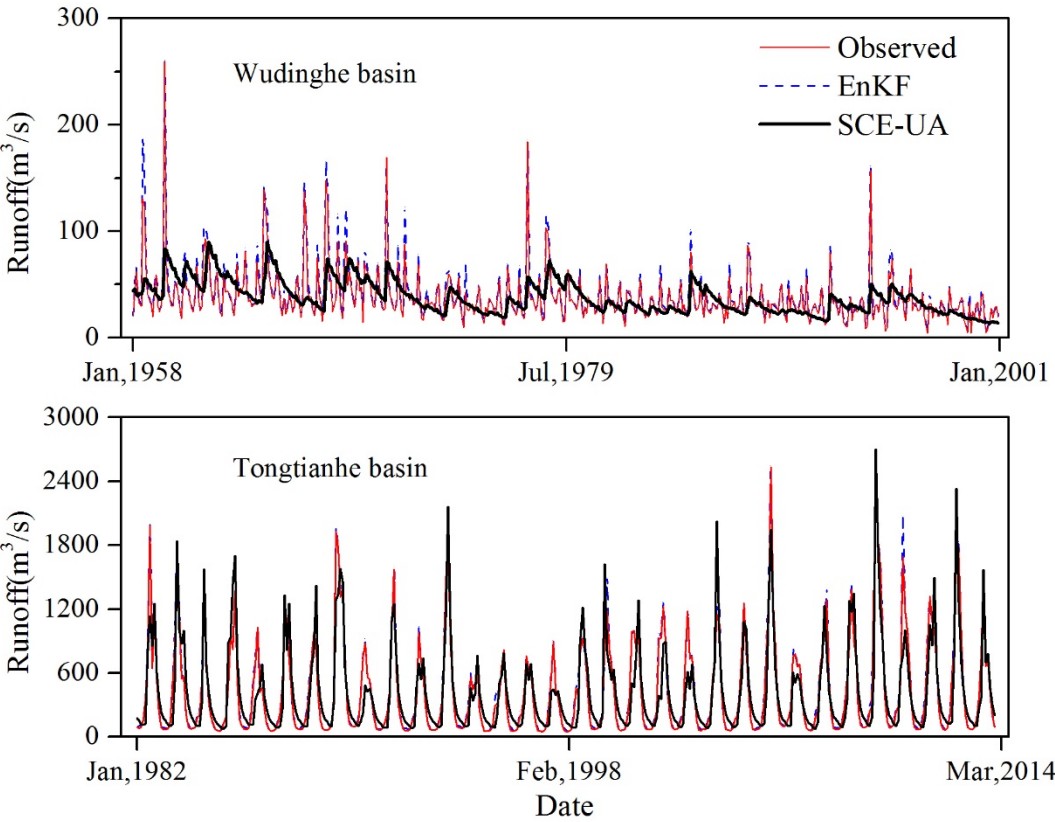

**Figure. 12.** Comparison of observed runoff and runoff estimations from the EnKF and SCE-UA for Wudinghe basin
within the period 1958-2000 (top figure) and Tongtianhe basin within the period 1982-2013 (bottom).