# Peer review of "Identification of hydrological model parameters variation using"

_Hydrology and Earth System Sciences, 2015_

## Referee Comment (RC1) · Anonymous Referee #1 · 24 Mar 2016

The authors present an approach to estimate time-variable model parameters within an Ensemble Kalman Filter based framework. Therefore, a two-parameter hydrological model is applied, which estimates runoff based on precipitation and evapotranspiration data. In contrast to other EnKF-applications with time-invariant model parameters, the state prediction is separated into a two-step process. First, the model parameters are predicted (with some well defined uncertainty bounds). The state is then predicted using these new parameters. By that, the presented method is able to describe and estimate temporal variations (e.g. trends) in model parameters.

First of all, I totally agree with the authors that the time-variability of model parameters require attention and are worth to analyze. This holds especially true in the context of a changing climate and anthropogenic interventions in the water budgets,

where relationships between variables and parameters might change significantly over time.

However, I was rather disappointed when reading through the paper. My main point of criticism is the overall immature state of the manuscript. While the title and the abstract sounded promising, the presented analyses, together with a significant lack of motivation, justification, and information left many open questions. The inconsistencies in the formulas as well as a confusing structure of the manuscript, bad language, and quite strange word choices further make it very difficult to understand what the authors really want to show and how they obtain their results. Overall, I have the feeling that the manuscript requires a thorough proof reading by e.g. an experienced senior-scientist.

Many parameters and variables seem to be defined completely arbitrarily or taken from other studies without motivating and discussing the reasons for these choices. Furthermore, there is no justification about the different methods applied (EnKF, the dual-state parameter estimation approach from Moradkhani, 2005, . . .). The authors simply take these methods as a given without discussing the advantages and disadvantages with respect to their study.

The results section left many open questions and lack of significant analyses and findings of the approach presented. The authors further draw some confusing conclusions from their data (e.g. the trend line in Fig. 10; increased water storage capacity in the basin due to "land use changes", but no trend in the estimated SC). It is further left open if the abrupt changes (Fig 10, top) and trends (Fig. 11) in the estimated parameters make sense and how they might influence the runoff estimates. In fact, I don't think that the comparison against the SCE-UA-method is reasonable for analyzing the strengths of the proposed method. The authors should rather compare the performance of the EnKF with and without time-variable model parameters. But in

the current version, it is completely unclear if the proposed method is able to improve the runoff estimates compared to a model run with time-invariant parameters.

Overall, this suggests that there is a lack of understanding and motivation behind the presented study and the methods applied. However, I think that the general idea of the development of a data driven (EnKF) and simple hydrological model (the two-parameter model) which considers time-variables parameters (the dual-state parameter estimation approach) sounds very interesting and is worth to investigate.

Therefore, I would be happy if the authors could improve the paper significantly, as their basic idea sounds really promising. But I am not sure if all the issues and inconsistencies in the paper can be addressed within major revisions. The authors might have to re-write large parts of the manuscript, add a lot more details and information, and perform new analyses and calculations. **Therefore, I have to reject a publication of the manuscript in its current form in HESS**.

Here are some further points which should be addressed in a potential revision:

1. You should add some more motivation of your work and a better overview over similar studies to the introduction.

2. Explain in more detail which methods you're using and why you're using them. Why do you use an EnKF? You don't have a lot of data so there is actually no need to approximate the propagated covariance matrices with the empirical sample covariances.

3. You refer multiple times to the term "data assimilation", which (in my opinion) does not make sense here. You are using only single time-series for precipitation, evapotranspiration, and runoff which are perturbed with some predefined noise. Thus, there is no real "assimilation" of e.g. a large ensemble of data into their

two-parameter model.

4. Go through your equations! You're using some parameters twice (e.g. S). Please explain how the "forward- and observation operator" look like. Furthermore, the definition of the variables seems quite confusing (e.g. below equation 9).

5. The set-up of the synthetic experiment needs to be discussed in more detail (Why did you chose these 4 scenarios? What do you want to find out with these? ...).

6. Combine all performance metrics (NSE, VE, RMSE, ...) in one section.

7. Explain in more detail how you generate the synthetic data and which observations you're using. Furthermore, it would make much more sense if you could use more data (e.g. different modeled and observed precipitation and evapotranspiration data). Then, you could derive some reasonable uncertainty bounds, which you could use within your EnKF.

8. What is the SCE-UA-method and why did you chose it for comparison?

9. You have to analyze your results more carefully. Please try to give an explanation for some of your findings (e.g. why are the results for the SC- better than the C-parameter and why is there a time lag in the "assimilated" C-parameter?).

10. The results from the case studies sound more like catchment description rather than a thorough analysis of the method, the estimated parameters, and the estimated runoff time-series.

11. You could combine Figures 1-4 and Figures 5-8 in two plots.

12. The distinction between the two trend lines in Fig. 10 (top) does not make a lot of sense. That being said, can you give an explanation for this sudden change in 1972?

---

## Referee Comment (RC2) · Anonymous Referee #2 · 31 May 2016

Dear Authors! This paper focuses on the identification of the variability and change of model parameters with over a long time period. A parsimonious rainfall-runoff model on a monthly time step with only two model parameters was used in this study. An EnKF approach is used to update the model parameters based on the observed runoff. This method is applied for a synthetic experiment and two case studies in China. The aim of the study is to show the capability of the EnKF approach to estimate the model parameters and their change over time. In my opinion this is a very promising and important issue and additional research in this field is important. Going through this specific paper about parameter estimation I was thinking that this is more a draft or concept version of a publication, than a paper ready for submission. The introduction and the comparison with other studies should be deeper than in this version. And the benefits of the specific EnKF approach used in this study are not clearly supported

by the results of the synthetic experiment and the two real case studies in China. In general my opinion about the scientific quality of this publication is in line with that of reviewer #1. A lot of additional work and analysis have to be included before this work should be published. I do not go very much into the details, but my major concerns are: -) The introduction and literature review should be extended – broader context -) The superior performance of the EnKF method was not obvious to me. A comparison with other sequential data assimilation techniques would have been helpful. At the other hand I didn't quite understand what the real benefit is – if the parameters are estimated from observed discharge data in the past, but the performance of the model is not tested for the "forecast" or "prediction" case when no runoff measurements are available. In my opinion this should be the most important indicator for the added value of the data assimilation routine. -) Looking at figure 11, it is really not clear to me how the data assimilation approach could help to estimate appropriate model parameters when the 95% uncertainty bounds are much smaller than annual and inter annual variations of the evapotranspiration parameter C. To summarize, I suggest major revisions before this paper should be published and the benefit of the proposed method can be presented to the international scientific community.

Best regards
* * *

---

## Author Comment (AC1) · 8 Jun 2016

**Response to Anonymous Referee #1**

**(1) The authors present an approach to estimate time-variable model parameters within an Ensemble Kalman Filter based framework. Therefore, a two-parameter hydrological model is applied, which estimates runoff based on precipitation and evapotranspiration data. In contrast to other EnKF-applications with time-invariant model parameters, the state prediction is separated into a two-step process. First, the model parameters are predicted (with some well defined uncertainty bounds). The state is then predicted using these new parameters. By that, the presented method is able to describe and estimate temporal variations (e.g. trends) in model parameters.**

**First of all, I totally agree with the authors that the time-variability of model parameters require attention and are worth to analyze. This holds especially true in the context of a changing climate and anthropogenic interventions in the water budgets, where relationships between variables and parameters might change significantly over time.**

**However, I was rather disappointed when reading through the paper. My main point of criticism is the overall immature state of the manuscript. While the title and the abstract sounded promising, the presented analyses, together with a significant lack of motivation, justification, and information left many open questions. The inconsistencies in the formulas as well as a confusing structure of the manuscript, bad language, and quite strange word choices further make it very difficult to understand what the authors really want to show and how they obtain their results. Overall, I have the feeling that the manuscript requires a thorough proof reading by e.g. an experienced senior-scientist.**

**Reply:**

We thank the Anonymous Referee #1 for the constructive comments and suggestions. All the comments are addressed below. The manuscript has been revised significantly by re-organizing the structure, adding more details and explanations for the motivation, method and results, as well as thorough proofreading. The formulas have

been updated, and the descriptions of variables have been clarified.

**(2) Many parameters and variables seem to be defined completely arbitrarily or taken from other studies without motivating and discussing the reasons for these choices. Furthermore, there is no justification about the different methods applied (EnKF, the dual-state parameter estimation approach from Moradkhani, 2005, . . .). The authors simply take these methods as a given without discussing the advantages and disadvantages with respect to their study.**

**Reply:**

The EnKF, which is a typical sequential data assimilation method, has been widely and successfully applied in hydrology (Abaza et al., 2014; DeChant and Moradkhani, 2014; Delijani et al., 2014; Tamura et al., 2014; Xue and Zhang, 2014; Deng et al., 2015) due to its applicability to a variety of nonlinear problems (Evensen, 2003; Weerts and El Serafy, 2006). Several studies have successfully used it to estimate model states and parameters (Moradkhani et al., 2005; Wang et al., 2009; Xie and Zhang, 2010; Xie and Zhang, 2013; Samuel et al., 2014).

There are two main methods for simultaneously estimating the states and parameters: (1) the dual state-parameter estimation method using two interactive filters (Moradkhani et al., 2005); and (2) the state augmentation method by adding parameters into the state vector (Wang et al., 2009). In this study, the state augmentation method is used since the parameter update equation of the EnKF is equivalent in the two methods.

More details about the justification and discussion of EnKF have been added in the revised manuscript. The advantages of the EnKF and its applications in time-invariant parameter estimations for hydrological models are added in Section 2.2 (Page 8-12, Line 116-182), as well as the related references. Methods for simultaneously estimating the states and parameters are discussed. Details about the error setting for states, parameters and observations are clarified. Different levels of observation uncertainty are added to examine the performance of the EnKF in the synthetic experiment.

**(3) The results section left many open questions and lack of significant analyses and findings of the approach presented. The authors further draw some confusing conclusions from their data (e.g. the trend line in Fig. 10; increased water storage capacity in the basin due to "land use changes", but no trend in the estimated SC). It is further left open if the abrupt changes (Fig 10, top) and trends (Fig. 11) in the estimated parameters make sense and how they might influence the runoff estimates. In fact, I don't think that the comparison against the SCE-UA-method is reasonable for analyzing the strengths of the proposed method. The authors should rather compare the performance of the EnKF with and without time-variable model parameters. But in the current version, it is completely unclear if the proposed method is able to improve the runoff estimates compared to a model run with time-invariant parameters.**

**Overall, this suggests that there is a lack of understanding and motivation behind the presented study and the methods applied. However, I think that the general idea of the development of a data driven (EnKF) and simple hydrological model (the two-parameter model) which considers time-variables parameters (the dual-state parameter estimation approach) sounds very interesting and is worth to investigate.**

**Therefore, I would be happy if the authors could improve the paper significantly, as their basic idea sounds really promising. But I am not sure if all the issues and inconsistencies in the paper can be addressed within major revisions. The authors might have to re-write large parts of the manuscript, add a lot more details and information, and perform new analyses and calculations. Therefore, I have to reject a publication of the manuscript in its current form in HESS.**

**Reply:**

The case studies include Wudinghe basin and Tongtianhe basin. The estimated *SC* for Wudinghe basin has an increasing trend; while there is no trend in the estimated *SC* for the Tongtianhe basin. The water storage capacity of the Wudinghe basin has been significantly affected by human activities for water and soil conservation. The step

change of *SC* occurred in 1972 (Figure 7c) corresponds to the abrupt change of double mass curve of monthly runoff and precipitation shown in the top panel of Figure 6, indicating the change of runoff coefficient due to the soil and water conservation measures. The abrupt change of runoff coefficient in Wudinghe basin has been reported in other studies (Li et al., 2014; Xu, 2011). The soil and water conservation measures contribute to the increased storage capacity and the declined runoff coefficient (data from Wang and Fan, 2003).

A comparison on model parameter identification is conducted by using Tongtianhe basin, where human activity is minimal. Since the physical characteristics related to the water storage capacity have no significant changes, the estimated *SC* has a stable value.

The comparison with the SCE-UA method is removed in the revised manuscript since this study is focused on identifying time-variant parameters but the SCE-UA is for estimating constant parameters. The scenario with time-invariant parameters (Scenario 5 in Table 3) has been added to evaluate the performance of EnKF. Besides, the observation uncertainty with various bounds is included in the revised manuscript (Section 3.1, Page 13-15, Line 209-241). Results are thoroughly analyzed in Section 4 (Page 17-22, Line 283-371). Details about the results from both basins are clarified, especially for the abrupt change of the estimated *SC*. The interpretations of the parameter trends are strengthened by adding related data and references.

**1. You should add some more motivation of your work and a better overview over similar studies to the introduction.**

**Reply:**

The objective of this paper is to identify temporal variation of model parameters and to interpret the variations through catchment properties. The data assimilation (DA) approach actually provides another method to identify the potential temporal variations of model parameters by updating them when observations are available (Liu and Gupta, 2007; Xie and Zhang, 2013). It has been successfully used to estimate model parameters in recent years (Moradkhani et al., 2005; Panzeri et al., 2013; Vrugt et al., 2013; Xie and Zhang, 2013; Shi et al., 2014; Xie et al, 2014). As a typical sequential DA method, EnKF is used to track the dynamics of the model parameters.

The Introduction section has been significantly modified in the revised manuscript (Section 1, Page 4-7, Line 37-90). The objective of this study is clarified. Details are added in Section 1, including the introductions of DA method and its hydrological applications, especially for model parameter estimation.

"The data assimilation (DA) actually provides another method to identify the potential temporal variations of model parameters by updating them in real-time when observations are available (Liu and Gupta, 2007; Xie and Zhang, 2013). The DA method has been widely applied in hydrology for soil moisture estimation (Han et al., 2012; Kumar et al., 2012) and flood forecasting (Liu et al., 2013; Abaza et al., 2014). It has also been successfully used to estimate model parameters (Moradkhani et al., 2005; Panzeri et al., 2013; Vrugt et al., 2013; Xie and Zhang, 2013; Shi et al., 2014; Xie et al, 2014). For example, Vrugt et al. (2013) proposed two types of Particle-DREAM method to track the evolving target distribution of HyMOD parameters, while the true parameters were assumed to be constant. Xie and Zhang (2013) used a partitioned forecast-update scheme based on the EnKF to retrive optimal parameters in a distributed hydrological model. Although the DA method has been used to estimate model parameters, these studies are focused on the estimation of constant parameters. Little attention has been paid to the identification of time-variant model parameters and the interpretation of their temporal variations based on the

climate conditions and/or catchment characteristics."

**2. Explain in more detail which methods you're using and why you're using them. Why do you use an EnKF? You don't have a lot of data so there is actually no need to approximate the propagated covariance matrices with the empirical sample covariances.**

**Reply:**

In this paper, EnKF is applied to a monthly water balance model for identifying temporal variations of model parameters. It is a typical sequential data assimilation method and has been successfully used for state and parameter estimations (Moradkhani et al., 2005; Wang et al., 2009; Xie and Zhang, 2010; Samuel et al., 2014). However, it has not been applied for identifying the temporal variation of model parameters. The EnKF framework is appropriate for estimating hydrological parameters based on the previous studies (Wang et al., 2009; Xie and Zhang, 2010; Samuel et al., 2014; Deng et al., 2015).

In EnKF, the state and observation equations are as follows:

$$\begin{pmatrix} \theta_{i+1|i}^k \\ x_{i+1|i}^k \end{pmatrix} = \begin{pmatrix} \theta_{i|i}^k \\ f\left(x_{i|i}^k, \theta_{i+1|i}^k, u_{i+1}\right) \end{pmatrix} + \begin{pmatrix} \delta_i^k \\ \varepsilon_i^k \end{pmatrix}, where\ \delta_i^k \sim N\left(0, U_i\right), \varepsilon_i^k \sim N\left(0, G_i\right) \qquad (1)$$

$$y_{i+1}^k = h\left(x_{i+1|i}^k, \theta_{i+1|i}^k\right) + \xi_{i+1}^k, \xi_{i+1}^k \sim N\left(0, W_{i+1}\right) \qquad (2)$$

where $x$ is the state vector with a dimension of $n \times 1$; $\theta$ is the parameter vector with a dimension of $l \times 1$. The augmented state vector includes both states and model parameters has a dimension of $(n+l) \times 1$. $\varepsilon_i$ and $\delta_i$ are the independent white noise for the state and parameter vector with a dimension of $n \times 1$, followed a Gaussian distribution with zero mean and covariance matrix $G_i$ and $U_i$ with a dimension of $n \times n$, respectively. $\xi_{i+1}$ is the noise term for the observation vector with a dimension of $m \times 1$ which follows a Gaussian distribution with zero mean and

covariance matrix $W_{i+1}$ with a dimension of $m \times m$.

Details added in Section 2.2 (Page 8-12, Line 116-182) include the application of EnKF in time-invariant parameter estimations for hydrological models, the method of updating state variables and parameters, and clarification of variables.

**3. You refer multiple times to the term "data assimilation", which (in my opinion) does not make sense here. You are using only single time-series for precipitation, evapotranspiration, and runoff which are perturbed with some predefined noise. Thus, there is no real "assimilation" of e.g. a large ensemble of data into their two-parameter model.**

**Reply:**

We agree with you that there is no large ensemble of data assimilated into the two-parameter monthly water balance model. But runoff observation is assimilated into the hydrologic model to estimate the model parameters. The similar hydrological applications of the DA method have been reported in previous studies (Wang et al., 2009; Xie and Zhang, 2010; Samuel et al., 2014).

**4. Go through your equations! You're using some parameters twice (e.g. S). Please explain how the "forward- and observation operator" look like. Furthermore, the definition of the variables seems quite confusing (e.g. below equation 9).**

**Reply:**

Thanks. The parameters S and R in Section 2.2 caused confusion with the soil water content S in the hydrological model and Pearson correlation coefficient R. R in Section 2.2 is changed to U; and S in Section 2.2 is changed to W.

The forward operator represents the state transfer function, and the observation operator represents the relationship between observations and states.

The description of the EnKF method including equations and introduction has been re-organized in Section 2.2 (Page 8-12, Line 116-182).

**5. The set-up of the synthetic experiment needs to be discussed in more detail (Why did you chose these 4 scenarios? What do you want to find out with these? . . .).**

**Reply:**

The true values of parameters are not available. The synthetic experiments are designed to evaluate the capability of EnKF to identify the temporal variations of model parameters. The scenarios for time-variant parameters are designed based on the changes of two parameters: (1) two cases of changes for parameter $C$, including periodic variation and periodic variation with an increasing trend, and (2) two cases of changes for parameter $SC$, including an increasing trend and abrupt change. Parameter $C$ is a parameter related to evapotranspiration, and the two cases are designed to represent the potential monthly and annual variations of actual evaporation. Parameter $SC$ is the catchment water storage capacity, and the two time series are designed to represent the change caused by human activities such as land use and land cover change.

The purpose and steps of the synthetic experiment are clarified. The reason for the setting of various parameter variations are explained in Section 3.1 (Page 13-14, Line 209-222).

"A synthetic experiment is designed to evaluate the capability of the assimilation procedure to identify the temporal variation of model parameters. Five scenarios of different parameter variations are developed, as shown in **Table 2**. The model parameters in the first four scenarios are time-variant, and those in the last scenario are constant. Parameter $C$, the evapotranspiration parameter, is considered to be sinusoidal reflecting potential seasonal variations in hydrological model parameters (Paik et al., 2005; Ye et al., 1997). An increasing trend is also considered to account for the potential annual or long-term variability. The change of parameter $SC$ is considered to be gradual and abrupt, since the catchment water storage capacity can be affected by land use and land cover changes, such as afforestation and dam construction. The parameters in Scenario 5 are treated as constants like the conventional hydrological modeling. Observations for precipitation and potential

evapotranspiration are generated by adding a Gaussian disturbance to the corresponding data from a real catchment, and runoff is then produced using the TWBM model. The data set used in this experiment includes a total of 672 months. The first 24-month period is set for model warm-up to reduce the impact of the initial soil moisture conditions."

**6. Combine all performance metrics (NSE, VE, RMSE, . . .) in one section.**

**Reply:**

Thanks. All the evaluation metrics have been combined in Section 2.3 (Page 12-13, Line 185-205).

**7. Explain in more detail how you generate the synthetic data and which observations you're using. Furthermore, it would make much more sense if you could use more data (e.g. different modeled and observed precipitation and evapotranspiration data). Then, you could derive some reasonable uncertainty bounds, which you could use within your EnKF.**

**Reply:**

The forcing data, i.e., precipitation and potential evapotranspiration, were generated via a stochastic simulation where a Gaussian noise was added to the data set from a real catchment. The time-variant and constant model parameters ($C$ and $SC$) are generated using a sinusoidal and/or linear function in their prior ranges (Table 1). The runoff series were obtained from the simulations of the TWBM model. The simulated runoff is considered as the observation to be assimilated. Different levels of observation uncertainty in the forcing data and runoff (Table 3) are added to examine the capability of the EnKF in the synthetic experiment.

Two sets of forcing data with different mean and variance are considered and similar results are obtained as those in the synthetic experiment. Figures A1 and A2 present the estimates for three scenarios under the low-level uncertainty in observation. The results show that the temporal variations of the true parameters can be successfully captured, although bias exists at the initial steps. Note that all the scenarios of

parameter variations are examined and similar results are obtained (not shown).

Details about the generation of the synthetic data are added in Section 3.1(Page 14-15, Line 223-230), as well as the different levels of observation uncertainty (Page 15, Line 236-241):

"(1) Time series of model parameters are generated, including the time-variant parameters and the constant parameters. Model parameter sets are produced using a sinusoidal function and/or a linear trend function within the specified ranges shown in **Table 1**. The runoff observations for each scenario are computed from the TWBM model taking monthly potential evapotranspiration and precipitation, and the parameters as inputs…

To evaluate the effect of errors on identifying parameter variation, different levels of observation uncertainty are considered in the synthetic experiment, as detailed in **Table 3**. The uncertainties from the observed precipitation and runoff are characterized by adding Gaussian noises where the standard deviations are assumed to be proportional to the magnitude of the true values, and the corresponding proportional factors are denoted as $\gamma_P$ and $\gamma_Q$. The proportional factors are set to account for the practical measurement error (Wang et al., 2009; Xie and Zhang, 2010)."

**Table 3.** Proportional factors of the standard deviations for precipitation ($\gamma_P$) and runoff ($\gamma_Q$) uncertainties.

| Type | Low level | Medium level | High level |
|---|---|---|---|
| $\gamma_P$ | 0 | 0.05 | 0.10 |
| $\gamma_Q$ | 0.05 | 0.10 | 0.20 |

[Figure]

**Figure A1.** Comparison between estimated *C* and its true value for three scenarios under different model inputs (i.e., precipitation and potential evapotranspiration). The grey areas represent the 95% uncertainty intervals. Note that Scenario 1: *C* has a periodic variation, and *SC* has an increasing trend; Scenario 2: *C* has a periodic variation, and *SC* has an abrupt change; Scenario 5: Both *C* and *SC* are constant.

[Figure]

**Figure A2.** Comparison between estimated *SC* and its true value for three scenarios under different model inputs (i.e., precipitation and potential evapotranspiration). The grey areas represent the 95% uncertainty intervals. Note that Scenario 1: *C* has a periodic variation, and SC has an increasing trend; Scenario 2: *C* has a periodic variation, and *SC* has an abrupt change; Scenario 5: Both *C* and *SC* are constant.

**8. What is the SCE-UA-method and why did you chose it for comparison?**

**Reply:**

The SCE-UA is a global optimization method and has been widely used for parameter estimation of hydrological models. The SCE-UA has been removed in the revised manuscript, since this study is focused on identifying the time-variant parameters while the SCE-UA is for estimating time-invariant parameters. To show the capability of the proposed method, the scenario of time-invariant parameters is added in Section 3.1 (Page 13-14, Line 210-212).

"Five scenarios of different parameter variations are developed, as shown in **Table 2**. The model parameters in the first four scenarios are time-variant, and those in the last scenario are constant."

Table 2. Different variations of model parameters in the synthetic experiment.

| Scenario | Description |
| --- | --- |
| Scenario 1 | $C$ has a periodic variation, and $SC$ has an increasing trend |
| Scenario 2 | $C$ has a periodic variation, and $SC$ has an abrupt change |
| Scenario 3 | $C$ has a periodic variation with an increasing trend, and $SC$ has an increasing trend |
| Scenario 4 | $C$ has a periodic variation with an increasing trend, and $SC$ has an abrupt change |
| Scenario 5 | Both $C$ and $SC$ are constant |

**9. You have to analyze your results more carefully. Please try to give an explanation for some of your findings (e.g. why are the results for the SC- better than the C-parameter and why is there a time lag in the "assimilated" C-parameter?).**

**Reply:**

The observation at the current time step is used to adjust the state variables and parameters in EnKF, and the updates of parameters depend on the Kalman gain for parameters. A runoff observation at the current time is determined by states at the current and previous time steps (Pauwels and Lannoy, 2006). The Kalman gain is dependent on the relative value of observation error to model error. The updated states are closer to the observation with a higher Kalman gain (Tamura et al., 2014). The $C$

series have higher coefficient of variation ($C_v$) value than the *SC* series in the synthetic experiment.

The synthetic *C* series were assumed to be periodic where lots of peak values exist; while the variation of *SC* series is less. The time lag between assimilated and true values exists especially when peak values occur (Clark et al., 2008; Samuel et al., 2014). Therefore, the *SC* with gradual and abrupt variations has better matches than the C with periodic variation.

The results have been modified in the revised manuscript. The analysis of estimated *SC* and *C* are extended in Section 4.1 (Page 18-19, Line 283-318). A more detailed explanation of the time lag is given as follows:

"It should be noted that there are time lags between the assimilated and true *C*. The observation at the current time step is used to adjust the state variables and parameters in EnKF, and the updates of parameters depend on the Kalman gain for parameters. A runoff observation at the current time is determined by states at the current and previous time steps (Pauwels and Lannoy, 2006). The Kalman gain is dependent on the relative value of observation error to model error. The updated states are closer to the observation with a higher Kalman gain (Tamura et al., 2014). The synthetic *C* series were assumed to be periodic where lots of peak values exist; while the variation of *SC* series is less. The time lag between assimilated and true values exists especially when peak values occur (Clark et al., 2008; Samuel et al., 2014)."

**10. The results from the case studies sound more like catchment description rather than a thorough analysis of the method, the estimated parameters, and the estimated runoff time-series.**

**Reply:**

The parameter *SC* represents the catchment water storage capacity and is related to the land use and land cover (LULC) of the catchment. The LULC in Wudinghe basin has been changed by soil and water conservation measures, such as tree and grass plantation and check dam construction, which play a significant role in strengthening the water holding capacity of the basin (Xu, 2011).

Part of the content has been moved to Section 3.2.1. The texts for results have been modified in the revised manuscript. The results of the estimated parameters are thoroughly analyzed in Section 4.2 (Page 20-22, Line 321-371). .

"The trend slopes of the two periods, one is from 1956 to 1971, the other is from 1972 to 2000, are different because the degree of implementing engineering measures varied during the period of 1958-2000. Moreover, the increase of the water holding capacity slowed down during the 1980s due to the sedimentation in reservoirs and check dams after periods of operation (Wang and Fan, 2003). **Fig. 8** shows the runoff reduction caused by all the soil and water conservation measures, i.e., land terracing, tree and grass plantation, check dam and reservoir construction. The runoff reduction positively relates to the water holding capacity, namely the *SC* value. The slope for the period of 1958-1971 is higher than that for the period of 1972-1996, suggesting that the *SC* in the former period has higher increasing trend. The runoff reduction data is available from 1956 to 1996 (Wang and Fan, 2003)…

For parameter *C*, the results show that the estimates have no obvious temporal patterns because the trend line slopes are almost zero and the standard deviations are relatively small for the two basins (**Fig. 7(a)** and **(b)**). However, the temporal variations exist in the estimated *C* values, indicating that this parameter has different values during the time steps and can be treated as time-variant parameters. The temporal variations of the estimated *C* are related to the variation of monthly actual evaporation, which is affected by multiple climatic factors, such as air temperature, soil moisture and solar irradiance (Su et al., 2015)…

The runoff simulations for both the two basins have good match with the runoff observations. Specifically, the *NSE* and *VE* for the Wudinghe basin are 0.93 and 0.07 respectively. While the corresponding index values are 0.99 and 0.04 for the Tongtianhe basin."

**11. You could combine Figures 1-4 and Figures 5-8 in two plots.**

**Reply:**

Thanks. The figures have been combined in the revised manuscript.

**12. The distinction between the two trend lines in Fig. 10 (top) does not make a lot of sense. That being said, can you give an explanation for this sudden change in 1972?**

**Reply:**

The abrupt change of *SC* corresponds to the change of the double mass curve for runoff and precipitation in the top panel of Figure 6. The double mass curve shows that a step change occurred in 1972, indicating that runoff coefficients had been changed since 1972 due to the soil and water conservation measures. The two trend lines are obtained based on the dividing point in Figure 6 (top). The abrupt change in runoff coefficient has been reported by Li et al. (2014) and Xu (2011). These measures increased the water storage capacity of the basin, explaining the increasing trend of parameter *SC* in Figure 7(c). The *SC* series of the two periods (pre- and post-1972) have different slopes for the trend lines, because the implementation of engineering measures have different degrees during the period of 1958-2000. Moreover, the increasing trend of water holding capacity slows down during the 1980s due to the sedimentation in reservoirs and check dams after periods of operation. Figure A3 shows the runoff reduction caused by all the soil and water conservation measures, i.e., land terracing, tree and grass plantation, check dam and reservoir construction. The runoff reduction positively relates to the water holding capacity, namely the *SC* value. The slope for the period of 1958-1971 is higher than that for the period of 1972-1996, suggesting that the *SC* in the former period has a higher increasing trend. Note that the runoff reduction data is available from 1956 to 1996.

This part has been modified in the revised manuscript (Section 4.2, Page 20-22, Line 321-371). In particular, the runoff reduction data from Wang and Fan (2003) are used to verify the abrupt change of *SC* series in the Wudinghe basin.

"The time series of estimated *SC* shows an apparent increasing trend, with two different trends for pre- and post-turning point in **Fig. 6(a)**. The temporal variation of the water storage capacity is correlated with the changes of land use and land cover.

Both the trends in **Fig. 7(c)** show an increase of *SC*, because the implementation of the large-scale engineering measures significantly improved the water holding capacity of the Wudinghe basin, especially for the reservoir and check dam construction. The trend slopes of the two periods, one is from 1956 to 1971, the other is from 1972 to 2000, are different because the degree of implementing engineering measures varied during the period of 1958-2000. Moreover, the increase of the water holding capacity slowed down during the 1980s due to the sedimentation in reservoirs and check dams after periods of operation (Wang and Fan, 2003). **Fig. 8** shows the runoff reduction caused by all the soil and water conservation measures, i.e., land terracing, tree and grass plantation, check dam and reservoir construction. The runoff reduction positively relates to the water holding capacity, namely the *SC* value. The slope for the period of 1958-1971 is higher than that for the period of 1972-1996, suggesting that the *SC* in the former period has higher increasing trend. The runoff reduction data is available from 1956 to 1996 (Wang and Fan, 2003)."

[Figure]

**Figure A3.** Runoff reduction caused by all the soil and water conservation measures, i.e., land terracing, tree and grass plantation, check dam and reservoir construction for the period of 1958 to 1996. The data is from Wang and Fan (2003).

---

## Author Comment (AC2) · 8 Jun 2016

**Response to Anonymous Referee #2**

**(1) Dear Authors! This paper focuses on the identification of the variability and change of model parameters with over a long time period. A parsimonious rainfall-runoff model on a monthly time step with only two model parameters was used in this study. An EnKF approach is used to update the model parameters based on the observed runoff. This method is applied for a synthetic experiment and two case studies in China. The aim of the study is to show the capability of the EnKF approach to estimate the model parameters and their change over time. In my opinion this is a very promising and important issue and additional research in this field is important.**

**Reply:**

We thank the Anonymous Referee #2 for the constructive comments and suggestions. All the comments have been responded below, and have been incorporated into the revised manuscript.

**(2) Going through this specific paper about parameter estimation I was thinking that this is more a draft or concept version of a publication, than a paper ready for submission. The introduction and the comparison with other studies should be deeper than in this version. And the benefits of the specific EnKF approach used in this study are not clearly supported by the results of the synthetic experiment and the two real case studies in China.**

**Reply:**

We have revised the manuscript thoroughly based on the constructive comments, especially for Introduction and Results sections. More details and explanations are added to clarify the motivation of this study. Details are as follows:

The results from the synthetic experiment demonstrate that the EnKF is able to detect the temporal trend of the true parameter values by updating the state variable and parameters based on the runoff observations, although a time lag exists when the parameter $C$ is periodic. The case study in Wudinghe basin aims to estimate the time

series of the model parameters and to provide an explanation for the parameter variation from the perspective of the physical characteristic changes. Meanwhile, a comparative study is implemented to investigate the variation of model parameters in the Tongtianhe basin, which is barely affected by human activities. The results from the Wudinghe basin show that the parameter of water storage capacity (*SC*) has a significant increasing trend for the period of 1958 to 2000, corresponding to the increase of the water holding capacity of the basin resulted from the implementation of the soil and water conservation measures, including land terracing, tree and grass plantation, and check dam and reservoir construction. While in the Tongtianhe basin, the parameter *SC* has no significant trend for the period of 1982-2013, which is consistent with the relatively stationary catchment characteristics. Therefore, the method proposed in this paper provides an effective tool for identifying time-variant model parameters for the two-parameter hydrologic model.

**(3) In general my opinion about the scientific quality of this publication is in line with that of reviewer #1. A lot of additional work and analysis have to be included before this work should be published.**

**Reply:**

The manuscript has been significantly revised based on the constructive and helpful comments from the reviewers. Please also refer to the responses to Reviewer #1.

**(4) I do not go very much into the details, but my major concerns are: The introduction and literature review should be extended – broader context.**

**Reply:**

Our paper aims to identify temporal variation of model parameters and to interpret the variations through catchment properties. Kalman filter and its extentions actually provide a method to identify the potential temporal variations of model parameters (Liu and Gupta, 2007; Xie and Zhang, 2013). Particularly, EnKF has been successfully applied in hydrology (Abaza et al., 2014; DeChant and Moradkhani, 2014; Delijani et al., 2014; Tamura et al., 2014; Xue and Zhang, 2014; Deng et al.,

2015). Several studies have used it to estimate model states and parameters under stationary conditions (Moradkhani et al., 2005; Wang et al., 2009; Xie and Zhang, 2010; Xie and Zhang, 2013; Samuel et al., 2014). In this study, we use the EnKF to detect the potential temporal variations of model parameters.

The introduction has been extended and more previous related references have been added in the revised manuscript (Page 4-6, Line 37-87).

"Parameters of conceptual hydrological models can be considered as a simplified representation of the physical characteristics in hydrologic processes. Therefore, parameter values are closely related to the catchment conditions, such as climate change, afforestation and urbanization (Peel et al., 2011)…

The data assimilation (DA) actually provides another method to identify the potential temporal variations of model parameters by updating them in real-time when observations are available (Liu and Gupta, 2007; Xie and Zhang, 2013). The DA method has been widely applied in hydrology for soil moisture estimation (Han et al., 2012; Kumar et al., 2012) and flood forecasting (Liu et al., 2013; Abaza et al., 2014). It has also been successfully used to estimate model parameters (Moradkhani et al., 2005; Panzeri et al., 2013; Vrugt et al., 2013; Xie and Zhang, 2013; Shi et al., 2014; Xie et al, 2014). For example, Vrugt et al. (2013) proposed two types of Particle-DREAM method to track the evolving target distribution of HyMOD parameters, while the true parameters were assumed to be constant. Xie and Zhang (2013) used a partitioned forecast-update scheme based on the EnKF to retrive optimal parameters in a distributed hydrological model. Although the DA method has been used to estimate model parameters, these studies are focused on the estimation of constant parameters. Little attention has been paid to the identification of time-variant model parameters and the interpretation of their temporal variations based on the climate conditions and/or catchment characteristics.

The aim of this study is to assess the capability of the DA method (i.e., the EnKF) to identify the temporal variations of the model parameters for a monthly water balance model. Thus, a synthetic experiment, including four scenarios with different parameter variations and one scenario with time-invariant parameters, is designed for

parameter estimation at different uncertainty levels. Furthermore, two case studies are implemented to estimate the model parameter series and to interpret the parameter variations in response to the changes in catchment characteristics, i.e., land use and land cover."

**(5) The superior performance of the EnKF method was not obvious to me. A comparison with other sequential data assimilation techniques would have been helpful. At the other hand I didn't quite understand what the real benefit is – if the parameters are estimated from observed discharge data in the past, but the performance of the model is not tested for the "forecast" or "prediction" case when no runoff measurements are available. In my opinion this should be the most important indicator for the added value of the data assimilation routine.**

**Reply:**

Although the DA method has been used to estimate model parameters, these studies focused on the estimation of constant parameters. Little attention has been paid to the identification of time-variant model parameters and the interpretation of their temporal variations based on the climate conditions and/or catchment characteristics. Therefore, EnKF is applied in this study for the estimation of model parameter series by using historical records of runoff. Afterwards, the parameter time series is analyzed for detecting potential temporal trend. The change of watershed characteristics are linked to the identified temporal variations of parameters. This study belongs to hydrological simulation and is an important step for achieving hydrological forecast. After the temporal pattern is identified and the impact factors are detected, we can build the function of time-variant parameters to predict the model parameters based on the factors, and further for hydrologic forecasts.

**(6) Looking at figure 11, it is really not clear to me how the data assimilation approach could help to estimate appropriate model parameters when the 95% uncertainty bounds are much smaller than annual and inter annual variations of the evapotranspiration parameter C.**

**Reply:**

Figure 11 shows the time series of the estimated *C* for the Wudinghe and Tongtianhe basins. Both the uncertainty bounds and the time series of parameters are obtained from the EnKF procedure. We focus more on the variability of the estimated parameter series, which cannot be obtained using the traditional way where optimal algorithm is applied for the estimation of the assumed constant parameter. The uncertainty bound is presented to show whether it can cover the true values of parameters in the synthetic experiment. However, the true parameters cannot be obtained in the real cases, it only shows the uncertainty level of the estimated parameters.

**(7) To summarize, I suggest major revisions before this paper should be published and the benefit of the proposed method can be presented to the international scientific community.**

**Reply:**

Thank you. We have revised the manuscript thoroughly based on the helpful comments, especially for the Introduction and Results sections, where more details and explanations are added to clarify the motivation of this study.